# Design and Test of Tread-Pattern Structure of Biomimetic Goat-Sole Tires

**DOI:** 10.3390/biomimetics7040236

**Published:** 2022-12-12

**Authors:** Fu Zhang, Yubo Qiu, Shuai Teng, Xiahua Cui, Xinyue Wang, Haoxun Sun, Shaukat Ali, Zhijun Guo, Jiajia Wang, Sanling Fu

**Affiliations:** 1College of Agricultural Equipment Engineering, Henan University of Science and Technology, Luoyang 471003, China; 2Collaborative Innovation Center of Machinery Equipment Advanced Manufacturing of Henan Province, Henan University of Science and Technology, Luoyang 471003, China; 3Luoyang Polytechnic, Luoyang 471000, China; 4Wah Engineering College, University of Wah, Wah Cantt 47040, Pakistan; 5College of Vehicle &Transportation Engineering, Henan University of Science and Technology, Luoyang 471003, China; 6College of Physical Engineering, Henan University of Science and Technology, Luoyang 471003, China

**Keywords:** bionic goat-hoof, tread pattern, adhesion, wheeled vehicles

## Abstract

To solve the technical problem that wheeled vehicles are prone to skidding on complex ground, due to poor adhesion performance, a tire-tread-structure design method based on the bionic principle is proposed in this paper. The 3D model of a goat’s foot was obtained using reverse engineering technology, and the curve equation was fitted by extracting the contour data of its outer-hoof flap edge, which was applied to the tire-tread-structure design. The bionic and herringbone-pattern rubber samples were manufactured, and a soil-tank test was carried out using an electronic universal tensile-testing machine, in order to verify the simulation results. The results showed that the overall adhesion of the bionic tread-pattern was greater than that of the normal tread-pattern with the same load applied and the same height and angle of the tread-pattern structure, and the maximum adhesion was increased by 14.23%. This research will provide a reference for optimizing the pattern structure and thus improving the passing performance of wheeled vehicles.

## 1. Introduction

Wheeled vehicles are simple in structure, highly mobile, and suitable for non-structural complex ground [1], but they are prone to skidding during travel, which greatly affects their operational efficiency. As the main component of wheeled vehicles in contact with the ground, the performance of the tires directly affects the passing performance of the vehicle [2,3,4].

The tire performance was improved by the shape, depth, arrangement angle and structure of the tread pattern. Wang et al. [5] used the VIC-3D non-contact strain-measurement system to extract the relevant deformation parameters of the contact region, using 10 types of 205/55R16 car radial (PCR) tires as the research object; the relationship between the identified deformation parameters and tire-performance indexes was established, and a tire-performance evaluation method was proposed, based on the deformation parameters. Based on the relationship between the above deformation parameters and tire-performance indexes, Liang et al. [6] established a finite element model of 205/55R16 complex-tread tire. In addition, the number and width of transverse grooves in the outer-shoulder area were simulated, to analyze the influence of the tire rolling-resistance and grip performance. The concave transverse-grooves with narrow grooves in the middle and wide grooves at the ends were optimized, and the test showed that the optimized tire rolling-resistance was reduced by 2.112 N and the grip force was increased by 10.196 N. Sridharan [7] studied the grounding-pressure-distribution characteristics of different-shaped tread blocks and their deformation characteristics when sliding in the contact area. It was found that the tread-block-structure form affected its grounding pressure and deformation characteristics, and reasonable control of the tread-block structure can effectively improve the tire grip-performance. Mundl et al. [8] showed that the tread-pattern structure affects the tread stiffness of the tire, thus changing the tire’s grip performance. Therefore, the optimization of the tread-pattern structure can effectively improve the tire grip-performance. Dolwichai et al. [9] conducted a theoretical analysis of the effect of the tread pattern groove-angle, groove-bottom radius of curvature and other parameters on tire-tread stiffness. Cho [10] investigated the effect of different tread depths and tread layouts on the traction performance of tires through snow-traction-performance tests, indicating that the tread-pattern structure is a key factor affecting the traction performance of tires. The three-dimensional groove-pattern technology developed by Nakajima et al. [11] realized shear self-locking during the deformation of the tread blocks; a torque that inhibits the shear deformation of the tread blocks was generated, and the effective contact-area was increased, thus realizing the improvement in tire grip-performance.

With the continuous improvement of tire-performance requirements, bionics was used to study the characteristics of animal feet, and the superior adhesion performance of animal feet was organically combined with tire-pattern structure, and the design of a bionic tire-pattern to improve tire performance gradually become a research hotspot [12,13]. Orndorf et al. [14] quantified the surface roughness of the paw pads by calculating the power spectral density of the surface roughness of the American black bear, brown bear, sun bear and polar bear. Based on the friction experiment with the surface of the 3D-printed model and the snow, it was found that the frictional shear stress of polar bear paw-pads on the snow increased by 1.3–1.5, times compared with other species, suggesting that taller papillae may compensate for frictional losses resulting from the relatively smaller paw-pads of polar bears compared with their close relatives. Clemente et al. [15] examined vertical ground-reaction forces, contact areas, and foot pressures for camels and alpacas of varying body masses as they moved at different walking speeds across the pressure plates. They also explored the extent to which fat pads may alleviate the foot pressures associated with increased body mass or movement speed. Zhang’s team [16,17,18] designed a Mars rover wheel based on the ostrich foot-contour, and the traction force of this wheel was increased by 8.6% compared with that of ordinary wheels. In addition, it was found that the sand fixation and flow-restriction effect of the bottom surface of the third toe was better, based on the discrete-element and finite-element analysis, and it was divided into three parts, including the forefoot gentle surface, middle-groove surface and heel convex-crown surface. Moreover, the bionic drum wheel and common drum wheel were prepared using nylon material. The test showed that the bionic wheel improved the maximum traction performance by 346%, compared with the common wheel. Ma et al. [19] obtained the point-cloud data of the third toe of the ostrich using a 3D scanner, based on their previous research on the surface of the third toe of the ostrich; the data were converted into a 3D model by Geomagic Studio (Geomagic, NC, USA), and imported into CATIA (Dassault, Paris, France) software to extract the contour curve of the third toe of the ostrich. Based on this, the bionic sand wheel was designed. The adhesion effect of the biomimetic sand-climbing wheel-surface was obviously higher than that of a conventional sand-climbing wheel-surface. Zhang [20] used reverse engineering technology to extract the contour line of the goat hoof-ball and analyzed the hoof-ball organization structure, designing the biomimetic goat-hoof track pattern by using the macroscopic features and microscopic characteristics of the goat hoof-ball. In addition, the adhesion performance of the imitation goat-hoof track pattern with that of the common single-line track pattern was compared. The test showed that the adhesion force of the bionic track-plate increased by 9.1%, compared with that of the common track-plate. Based on the principle of suction-cup suction on an animal octopus-claw, Zhou [21] proposed a suction-cup bionic pattern anti-skid tire as a way of improving the adhesion of the tire on an icy road. Li et al. [22,23,24] developed desert bionic-tires in accordance with the structure and shape of a camel’s foot and its walking gait, which can adapt to the desert environment. The comparison between a desert bionic-tire and an ordinary tire showed that desert bionic-tires have nonlinear characteristics of elastic force, and the traction was significantly better than ordinary rubber tires when driving in the sand.

To sum up, scholars in the past considered the tire driving on a hard road surface, and improved the performance of the tire by designing the tire-pattern shape, depth, arrangement angle and tread-structure. Some scholars have introduced bionic structure into tire design to improve tire performance on a wet road, on Mars, in the desert, and in other conditions. However, there were few reports on the design of the tire pattern based on a goat’s foot characteristics to improve the adhesion of tire on unstructured road surface.

The team’s previous research found that the irregular shape of the goat’s foot surface could adapt to different ground and had a strong adhesion ability [25,26,27,28,29]. Therefore, in this study, the goat-sole were used as the bionic prototype, and the foot features were extracted, based on the bionic engineering principle, to carry out the study on the adhesion performance of tire tread-pattern structure of biomimetic goat-sole, which can effectively solve the problem of easy slippage during the walking process and provide an important reference for improving the passing performance of wheeled vehicles.

## 2. Bionic Tire-Tread Design

The goat breed used in this research was the Boer goat, aged approximately 6 months, and an adult sheep with strong and powerful limbs, no known foot pathologies, intact foot contours, and healthy body. This met the test requirements, and the test goat is shown in Figure 1a. The goat was scanned with SOMATOM Definition AS X-ray computed tomography equipment (Siemens, Berlin, Germany) (voltage 140 kV, current 240 mA, layer thickness 0.6 mm). The data was processed using the medical imaging software Mimics (Materialise, Louvain, Belgium), to obtain a rough 3D model, which was then imported into 3-matic software (Materialise, Louvain, Belgium) to complete the operations of filtering triangles, smoothing, and clipping, to obtain a 3D model that conformed to the later mathematical modeling, as shown in Figure 1b.

After the 3D-model bone-density analysis, it was found that the hind legs of the goats produced more thrust during movement. In accordance with the principle of symmetry, there was no difference between the left and right hind foot of the selected goat, and the external hoof-flap of the goat was more robust than the internal hoof-flap. In the research, the external hoof-flap of the left hindfoot was selected as the bionic prototype, as shown in Figure 2a. CATIA (Dassault, Paris, France) software was used to extract the point-cloud data of the left hindfoot, and the chordal deviation method was adopted to simplify the point-cloud data, and then the point-cloud of the sole-edge contour was extracted, as shown in Figure 2b. The *x* and *y* values of the plantar-contour point-cloud were imported into Matlab (MathWorks, Natick, MA, USA) software, and a 3rd order polynomial fit was selected to fit the data. The fitting results are shown in Table 1.

After extracting the edge curve of the goat-sole, the inner- and outer-contour curves of the external-foot-flap of the left hind-foot were selected to fit the equation-driven curve as the left and right contour-curves of the bionic unit. Considering the size of the actual tire-pattern, the upper and lower arcs of R12 and R24 were selected to be tangent to the bionic profiles on both sides to form a closed-contour curve, then the bionic-unit structure was formed by stretching, as in Figure 3.

The center of the upper and lower arcs of the bionic unit was used to establish a straight line, and the angle between the straight line and the substrate was regarded as the angle between the bionic-unit structure and the substrate. According to the standard of the ordinary herringbone-angle [30], four kinds of bionic tire-block with angles of 30°, as shown in Figure 4a, or 40° and pattern heights of 15 mm or 25 mm were designed. The surface area, angle, height and distance of the herringbone pattern in the comparative test were consistent with those of the bionic tire-block. The feasibility of the bionic pattern-design was verified by the simulation and experiment.

## 3. Simulation Test

The bionic tire-tread pattern and the common herringbone-tread pattern were simulated and analyzed with the ground shear-force under the same load based on the ABAQUS (Dassault, Paris, France) analysis solver. The bionic tire-tread pattern included the tread height and angle with the substrate, and the bionic tire-tread pattern with the best combination of tread height and substrate angle was derived, and then compared with the common herringbone-tread-pattern for analysis. The simulation scheme is shown in Table 2.

CATIA (Dassault, Paris, France) software was used to establish the soil-bed model of 810 mm × 620 mm × 115 mm and the six tire-block models of 300 mm × 300 mm × 20 mm. The specific pattern parameters are shown in Table 2. The models were imported into ABAQUS (Dassault, Paris, France) and processed, as shown in Figure 5.

Material properties were obtained through measurement tests: the internal friction angle and cohesion of the soil were measured using the shear instrument (ZQB-4, Nanjing Soil Instrument Factory, Nanjing, China). The soil elastic-modulus and Poisson’s ratio were measured using the universal testing machine (DNS02, Changchun Institute of Mechanical Science Co., LTD., Changchun, China). The soil density was measured using the ring-knife method. The rubber hardness was measured using the rubber-hardness tester (LX-A, Yueqing Aidebao Instrument Co., LTD., Yueqing, China).

The Mohr–Coulomb model was adopted for the soil model.
(1)τ=c+σtanφ 
in which τ is the shear strength of the soil, the unit is kPa, c is soil cohesion, the unit is kPa, σ is the positive pressure on the shear plane, the unit is kPa, φ is the angle of internal friction of the soil, and the unit is °.

Through measurement and calculation, the soil density was 1330 kg/m^3^, the elastic modulus was 10 MPa, the Poisson’s ratio was 0.33 [20], the internal friction angle was 28.85°, and the cohesion was 44.08 kPa.

The Mooney–Rivlin constitutive model was adopted for the rubber material.
(2)$$E=\frac{15.75+2.15HA}{100-HA}\;C_{10}=\frac{E}{6}\;C_{01}=0.25C_{10}$$
in which E is the elasticity modulus of rubber, the unit is MPa, HA is the shore hardness of rubber, the unit is *HA*, $C_{10}$ and $C_{01}$ are Poisson’s ratio, and they are positive-definite constants.

Through measurement and calculation, the rubber elastic-modulus was 4.44 MPa. Poisson’s ratio was 0.47. The model coefficient, $C_{10}$, was 0.74, and $C_{01}$ was 0.19.

The cell type of both was C3D8R when dividing the grid. For the soil model, the bottom surface was fully constrained. The forward speed of the tire sample was 8 mm/s, and the direction was parallel to the soil bed along the *y*-axis. The workstation used for performing the simulations was the Dell Precision 7820 Tower (CPU model: Xeon Silver 4210R, CPU frequency: 2.4 GHz, RAM: 32G) (Dell, TX, USA).

After finishing the setting, the job module was entered to analyze the finite element simulation of four kinds of bionic patterns and two kinds of common patterns, and the displacement–adhesion curves were output, as shown in Figure 6.

When the tire-block slid relative to the ground, the adhesion force mainly came from the shear force between the tire block touching the ground vertically and the soil and the friction force between the surface of the tire block and the soil. Therefore, the simulation-output traction force was the tire adhesion-force. As can be seen from Figure 6, the change trend of six kinds of tire-block adhesion force was basically the same; in the range of displacement 0~20 mm, the traction-force value had an obvious rising trend, which was the elastic behavior. After 60 mm, the traction-force value tended to be stable. Due to the phenomenon of congestion, the traction force gradually increased. In order to ensure the accuracy of the test results, the elastic-behavior data and the rear-congestion data at the beginning of the test were excluded, while the traction force in the displacement range of 60~140 mm was selected for analysis. Finite element simulation showed that under the same pattern height, angle and load conditions, the adhesion of the bionic pattern was generally better than that of the ordinary pattern, with the maximum adhesion increase of 14.23%. In the case of the same pattern, the pattern angle was 30°, the height was 25 mm, and the adhesion force was larger. The bionic pattern had the best adhesion performance when the angle was 30° and the height was 25 mm (i.e., for pattern C).

## 4. Soil Tank Test

To verify the feasibility of the bionic-tread design and the accuracy of the simulation-test results, the rubber sheet was processed with a CNC engraving machine (Classic-103, MultiCam Inc., Dallas, TX, USA) to prepare a test sample as shown in Figure 7, and a tire-tread test system was built as shown in Figure 8, to complete the soil-groove test.

The test soil-tank size was 100 cm × 70 cm × 60 cm, and traction speed was 8 mm/s. Through the electronic universal tensile-tester traction around the fixed pulley wire-rope to achieve horizontal movement, the resistance signal was converted into an electrical signal by the tensile sensor and then transmitted to the tensile-tester data-acquisition-box, and then input into the computer records, display and post-processing.

The soil was turned deeply and sprinkled with water and other pretreatments to maintain the soil moisture-content of 10 ± 0.2% before the test. The soil was fully loosened, beaten well, and then scraped with a scraper and compacted with a roller. The above measures were taken to ensure that the mechanical properties of the soil in each test were consistent, so as to enhance the feasibility ratio of the test data [31].

As shown in Figure 9, each tire block was moved in the direction shown, as traction with a load of 100 N. To prevent boundary effects, each traction distance was 200 mm and each tread was tested five times, to ensure the validity of the test.

The soil-tank test data were imported into Origin (OriginLab, Northampton, MA, USA) software to plot the displacement–attachment force variation, as shown in Figure 10. The study showed that the test curve was consistent with the simulation-curve variation pattern, and the finite-element-analysis results effectively reflected the force process of the relative motion of the tire block and the soil.

The test data were compared with the finite-element-simulation data through 30 repetitive soil-tank tests with traction force as the verification index, as shown in Table 3. By comparing and analyzing the relative errors of the test and simulation traction-force, the maximum error was 18.17% and the minimum error was 10.22%, which proved that the finite-element-simulation results met the accuracy requirements. The error was caused by the following reasons: First, the soil was idealized in the simulation, while the soil environment was complex in the actual experiment. Second, although the physical properties of the soil tank were ensured to be in strict accordance with the test scheme each time, there are still human errors.

## 5. Conclusions

In this research, in order to solve the technical problems of wheeled vehicles that are prone to skidding on complex road surfaces, a study on the adhesion performance of the tire tread-pattern structure of biomimetic goat-sole was carried out, and the bionic tire-tread pattern was prepared, based on the bionic-engineering principle, and then the simulation and test-results verification was completed.

The curve of goat’s foot-flap profile was extracted, its mathematical model was established, and it was used to design the tire-tread pattern monolith that imitated the shape of the goat’s foot.

Finite element simulation showed that under the same pattern height, angle and load conditions, the adhesion of the bionic pattern was generally better than that of the ordinary pattern. The comparison between the results of the tire-pattern simulation and soil-groove test showed that the traction forces of the simulation and soil-groove test had the same upward trend, and the maximum error of the average traction-forces of the two was 18.17%, which verified the feasibility of the simulation results and the bionic tire-pattern, and provides a new research idea for tire-pattern design.

At present, only the static characteristics of a single foot-flap were studied in this paper, and the motor excellence of goats also benefits from the dynamic coordination of the two foot lobes. The relationship between the angle of bipedal flap and the adhesion performance of goats was further studied by using the high-speed camera system and QTM motion-capture system.

## Figures and Tables

**Figure 1 biomimetics-07-00236-f001:**
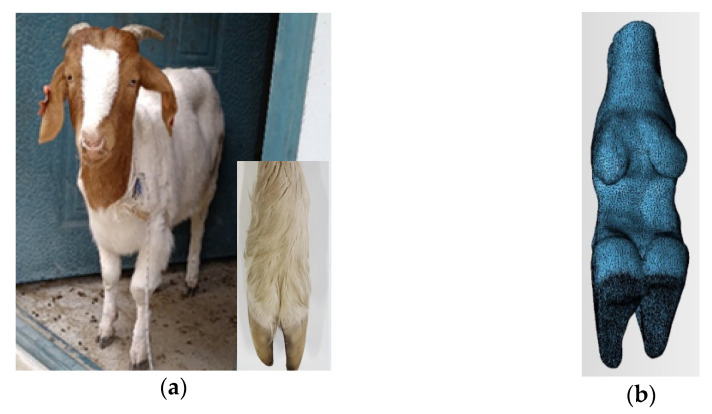
Test goat with goat-foot model: (**a**) Test goat; (**b**) 3D-model of goat foot.

**Figure 2 biomimetics-07-00236-f002:**
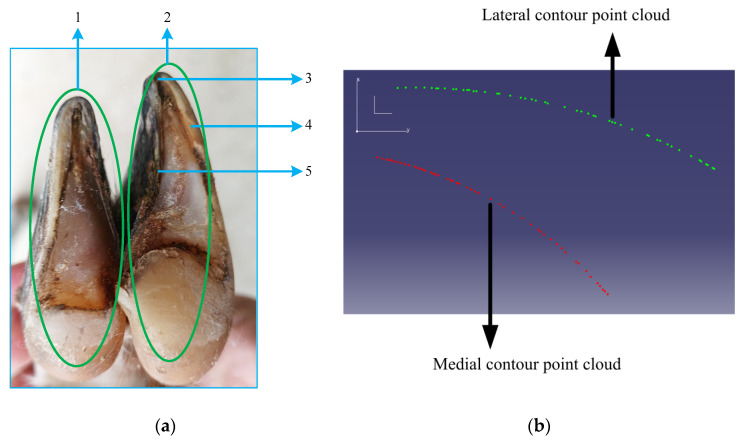
Point-cloud model of goat foot and edge contour: (**a**) left hind-foot of goat consists of 1. internal pedicle, 2. external pedicle, 3. toe, 4. lateral contour, 5. medial contour; (**b**) point cloud of external-foot flap-edge contour.

**Figure 3 biomimetics-07-00236-f003:**
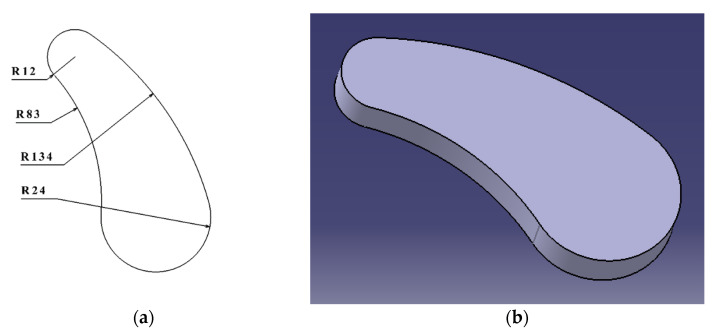
Bionic-pattern model: (**a**) schematic diagram of the bionic unit, the unit is mm; (**b**) structural diagram of the bionic unit.

**Figure 4 biomimetics-07-00236-f004:**
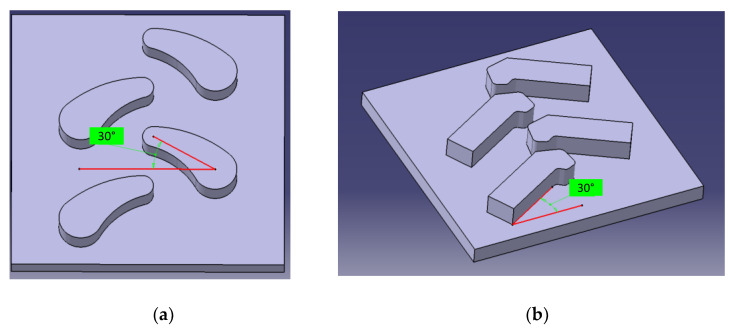
Tire block 3D-model: (**a**) biomimetic tire-block-model; (**b**) Common tire-block-model.0.

**Figure 5 biomimetics-07-00236-f005:**
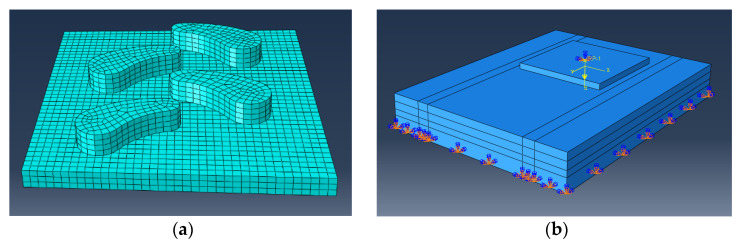
Simulation model: (**a**) the bionic-tire-block model after grid division; (**b**) the simulation model with constraints set. (note: in the figure, RP-1 is the reference point of vector force, which is equivalent to applying a certain load at the center of mass of the bionic tire-block. The orange and blue symbol at the bottom is a constraint on the soil bed which prohibits it from moving).

**Figure 6 biomimetics-07-00236-f006:**
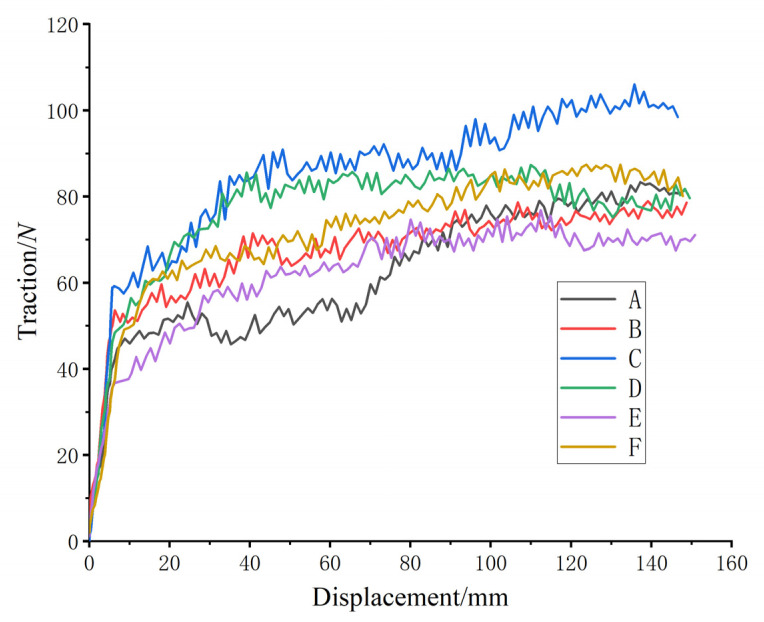
Adhesion output-curves between each tire pattern and soil. (A to F in Figure 6 correspond to A–F in Table 2).

**Figure 7 biomimetics-07-00236-f007:**
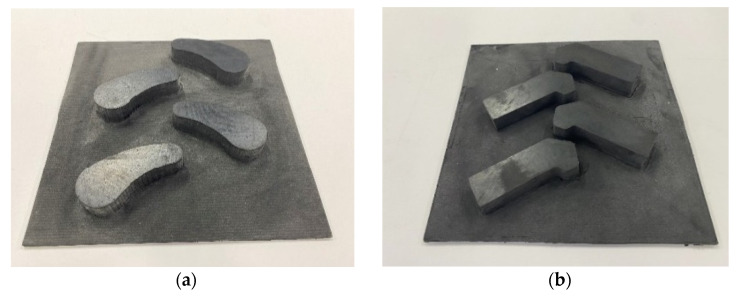
Tire block specimen: (**a**) bionic tire-block specimen I; (**b**) ordinary tire-block specimen I.

**Figure 8 biomimetics-07-00236-f008:**
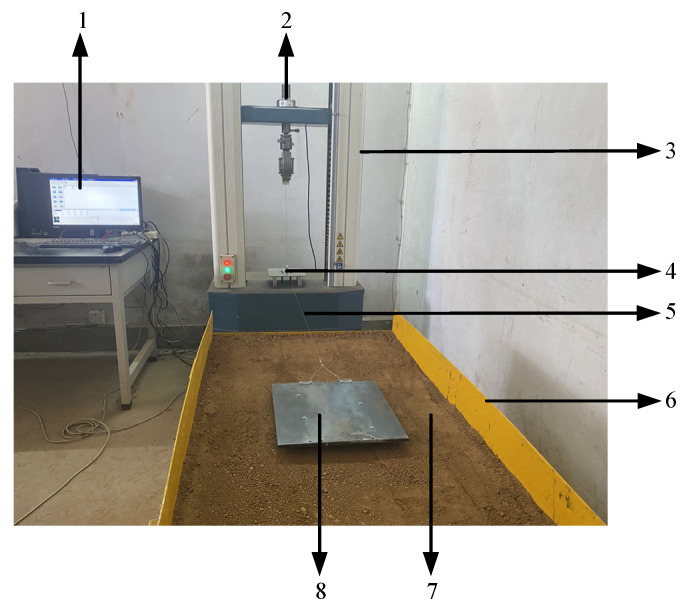
Tire-pattern test system consists of: 1. Computer, 2. Tension sensor, 3. Electronic universal tensile-tester, 4. Fixed pulley, 5. Traction wire-rope. 6. Soil groove, 7. Soil, 8. Bionic pattern.

**Figure 9 biomimetics-07-00236-f009:**
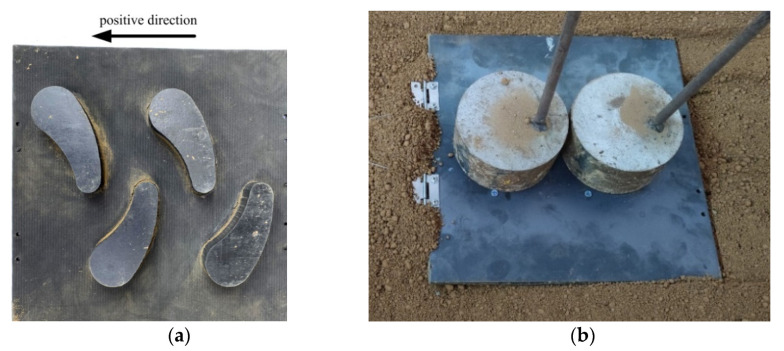
Test direction and load: (**a**) pattern direction; (**b**) apply 100 N load.

**Figure 10 biomimetics-07-00236-f010:**
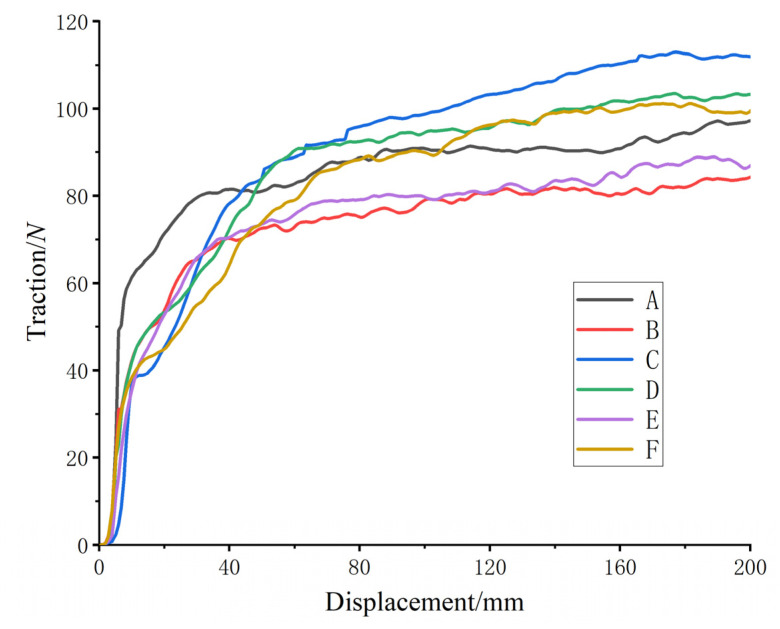
Tire tread-to-soil adhesion output-curve for each tire-groove test.

**Table 1 biomimetics-07-00236-t001:** External-foot flap-edge profile-curve fitting.

Result Parameters	External-Foot-Flap Lateral-Curve Fitting	External-Foot-Flap Medial-Curve Fitting
Functional equation	$y=-4.244e^{-5}x^{3}-0.005898x^{2}+0.06417x+4.524$	$y=-0.0002776x^{3}-0.005863x^{2}-0.2085x-2.254$
RMSE	0.018	0.033
SSE	0.013	0.043
Adjusted R-square	0.997	0.997
R-Square (R^2^)	0.996	0.996

**Table 2 biomimetics-07-00236-t002:** Simulation Scheme.

Number	Type	Height/mm	Clamping Angle/°	Load/N
A	Bionic pattern	15	30	100
B		15	40	100
C		25	30	100
D		25	40	100
E	Herringbone Pattern	15	30	100
F		25	30	100

**Table 3 biomimetics-07-00236-t003:** Comparison table of tested and simulated adhesion.

Number	Test Adhesion/N	Simulation Adhesion/N	Error
A	91.99	78.97	16.48%
B	81.87	73.90	10.78%
C	110.50	100.25	10.22%
D	94.60	80.05	18.17%
E	82.87	70.83	16.99%
F	98.30	85.37	15.14%

## Data Availability

Not applicable.

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
