# Peer review of "Design and Test of Tread-Pattern Structure of Biomimetic Goat-Sole Tires"

_biomimetics, 2022, doi:10.3390/biomimetics7040236_

Round 1
Reviewer 1 Report
Table 1: Results Parameters change in parameters. It is not needed to provide four decimals. Two are sufficient.
Line 135…“From the fitting results, it can be seen that Adjusted R2 and R2 are close to 1…”: not a result. Delete
Line 144. Explain R12,… in the legend
Figure 4 must be quoted in the text of the results. This is one example (if I am not wrong) with an angle of 30 degrees.
I do not understand why ° (Table 2) and deg (Figure 4) are used for describing (I think) the same unit. This unit should be consistently used throughout the MS.
Table 2: what is No? Explain or use another term
Figure 5: the legend should explain what we can see in the figure. It should be easier for the reader (e.g., biologist) to have a clear understanding of the figure. For example: what are the small 3D "digits"?
Figure 6: indicate that A to F corresponds to A – F in Table 2
Suggestion: Different terms are used to describe similar structures: ordinary tire (L. 212) and tire block (ordinary?) It should be better to use similar terminology throughout the text. I.e., ordinary tire block
I think that Methods and Results are put together. This is useful to understand the logic of the paper. But, why the animals are observed on a different surface than the tire? In figure 1, the goat is moving on an inclined surface and the tires are tested on a horizontal surface (figure 8). These surface physical properties should be consistent throughout the text. Can you explain or justify?
Conclusion: rather short. It seems to be an abstract of the methods (L 244 – 252). Then the conclusion arrives…The authors should discuss their results in a more general approach. Are there other studies of tire patterns on the basis of other animal structures? If yes, the methods are similar? It is possible to compare it with other data? Etc.
See useful recent paper (I think not cited):
Clemente, C.J., Dick, T.J.M., Glen, C.L. et al. Biomechanical insights into the role of foot pads during locomotion in camelid species. Sci Rep 10, 3856 (2020). https://doi-org.inee.bib.cnrs.fr/10.1038/s41598-020-60795-9
Ivanović, L., Vencl, A., Stojanović, B., & Marković, B. (2018). Biomimetics design for tribological applications. Tribology in Industry, 40(3), 448-456.
Orndorf, N., Garner, A. M., & Dhinojwala, A. (2022). Polar bear paw pad surface roughness and its relevance to contact mechanics on snow. Journal of the Royal Society Interface, 19(196), 20220466.
Author Response
Dear teacher,
Thank you for your hard work on the paper " Design and Test of Tread Pattern Structure of Biomimetic Goat Sole Tires " (ID: biomimetics-2055927), and thank you for valuable comments on the paper! This has significance instruction for our future research work. Now we make the following reply to your comments as the attachment.
Thank you very much.
With kind regards,
Fu Zhang

Reviewer 2 Report
In this manuscript, the authors have designed and evaluated a new tire tread inspired from goat`s foot to increase tire adhesion and prevent skidding of wheeled vehicles. The new tread and existing tread models were evaluated both numerically using a finite element model and experimentally and it was found the new tread contributes to higher adhesion. After reading the manuscript the reviewer, suggests major revision of the manuscript by addressing the following comments:
1. Introduction: a good background was presented with summary from the literature. However, the limitations and gaps from the past studies not much discussed and it is expected from the authors to provide a critical review of literature in this section. This section needs a revision. The research gap mentioned very briefly and could be elaborated further. Other comments:
Page 1, line 39 what tire parameters do affect the tire performance, must be stated more clearly
Page 2 line 64, “Bridgestone, Japan” should be cited appropriately. [similar to page 3, line 115]
Page 2 line 82, “Geomagic Studio” and “CATIA” should be cited appropriately, [similar to page 3, line 115]
Page 3, line 110, has the test goat reached maturity, or the foot contours are subjected to change? This should be elaborated.
Page 3 line 125 and 128, “CATIA” and MATLAB should be cited appropriately, [similar to page 3, line 115]
Figure 3, dimension units are missing (could be stated in the figure caption)
Page 5 line 125 and 158, “ABAQUS” should be cited appropriately, [similar to page 3, line 115]
Page 5 lines 165-170 could the author clarify how the material properties obtained? [from literature or were measured] What is model coefficient?
Page 6 line 173 could author provide more information on element size of the tire tread geometry and soil model as well as total elements of each geometry? Also the hardware information (CPU, RAM, etc) for the computer which performed the simulations.
Page 6 line 170, element type C3D8R was used indicating modelling soil as a single phase material, however in the soil tank test soil moisture content was stated, could authors justify why a biphasic model not adopted for soil model?
Page 6 line 174, is “A” a typo here?
Page 6 , line 190 could authors clarify what was requested from ABAQUS to plot results we see in Figure 6? Also it is worth to include any relevant formula or equations
Page 8, line 227, what does “positive traction direction of the bionic pattern” mean?
Page 8, lines 240-242 could authors discuss the reasons behind errors they observed between numerical results and experimental results?
Page 9 line 254, the 14.23% increase in adhesion as a result of new tread pattern not discussed in the main body of manuscript and only stated in abstract and conclusion. Similarly, pattern angle and height stated in the next sentence. Could authors cover these findings in the result section, as the conclusion should summarise the main points from the manuscript.
Discussion: The reviewer suggests that the authors discuss the effect of increased adhesion on friction coefficient and and potential increased wear of tires.
Also, what are the limitations of this study and what can be studied in future? Could authors clarify these in the conclusion ?
Author Response

(The authors gave the same response as above.)
